

# First insight into the viral community of the cnidarian model metaorganism Aiptasia using RNA-Seq data

Jan D. Brüwer and Christian R. Voolstra

Red Sea Research Center, Division of Biological and Environmental Science and Engineering (BESE), King Abdullah University of Science and Technology (KAUST), Thuwal, Makkah, Saudi Arabia

## ABSTRACT

Current research posits that all multicellular organisms live in symbioses with associated microorganisms and form so-called metaorganisms or holobionts. Cnidarian metaorganisms are of specific interest given that stony corals provide the foundation of the globally threatened coral reef ecosystems. To gain first insight into viruses associated with the coral model system Aiptasia (*sensu Exaiptasia pallida*), we analyzed an existing RNA-Seq dataset of aposymbiotic, partially populated, and fully symbiotic Aiptasia CC7 anemones with *Symbiodinium*. Our approach included the selective removal of anemone host and algal endosymbiont sequences and subsequent microbial sequence annotation. Of a total of 297 million raw sequence reads, 8.6 million (~3%) remained after host and endosymbiont sequence removal. Of these, 3,293 sequences could be assigned as of viral origin. Taxonomic annotation of these sequences suggests that Aiptasia is associated with a diverse viral community, comprising 116 viral taxa covering 40 families. The viral assemblage was dominated by viruses from the families *Herpesviridae* (12.00%), *Partitiviridae* (9.93%), and *Picornaviridae* (9.87%). Despite an overall stable viral assemblage, we found that some viral taxa exhibited significant changes in their relative abundance when Aiptasia engaged in a symbiotic relationship with *Symbiodinium*. Elucidation of viral taxa consistently present across all conditions revealed a core virome of 15 viral taxa from 11 viral families, encompassing many viruses previously reported as members of coral viromes. Despite the non-random selection of viral genetic material due to the nature of the sequencing data analyzed, our study provides a first insight into the viral community associated with Aiptasia. Similarities of the Aiptasia viral community with those of corals corroborate the application of Aiptasia as a model system to study coral holobionts. Further, the change in abundance of certain viral taxa across different symbiotic states suggests a role of viruses in the algal endosymbiosis, but the functional significance of this remains to be determined.

## INTRODUCTION

Research in the last few decades supports the notion that multicellular organisms do not live in isolation, but form complex associations with a variety of microorganisms including bacteria, archaea, and viruses (*McFall-Ngai et al., 2013*). The entity of host

Corresponding author
Christian R. Voolstra,
christian.voolstra@kaust.edu.sa

organism and microorganisms is termed 'metaorganism' or 'holobiont' (*Rohwer et al., 2002*; *Knowlton & Rohwer, 2003*; *Bosch & McFall-Ngai, 2011*). Among invertebrate animal hosts, stony corals form holobionts of particular interest given they engage in endosymbioses with photosynthetic algae of the genus *Symbiodinium* that form the basis of coral reef ecosystems (*Muscatine & Porter, 1977*; *Hoegh-Guldberg, 1999*). While the cnidarian host provides a light-rich and sheltered environment, *Symbiodinium* supply energy-rich sugars in the form of photosynthates (*Muscatine, 1967*; *Falkowski et al., 1984*). In addition, associated bacteria provide functions important for nutrient cycling (*Lesser & Jarett, 2014*; *Rädecker et al., 2015*), pathogen defense through the production of antibiotics (providing a function related to immunity), and potentially stress resilience (*Rosenberg et al., 2007*; *Shnit-Orland, Sivan & Kushmaro, 2012*; *Torda et al., 2017*; *Ziegler et al., 2017*). More recently, the importance of the viral community has shifted into focus. While the functional importance of coral-associated viruses is not entirely clear, recent studies suggest that viruses play a role in some coral diseases and potentially coral bleaching (*Marhaver, Edwards & Rohwer, 2008*; *Soffer et al., 2014*; *Weynberg et al., 2015*; *Weynberg et al., 2017b*; *Correa et al., 2016*; *Brüwer et al., 2017*; *Vega Thurber et al., 2017*; *Levin et al., 2017*).

Corals are under increasing threat from anthropogenic influences, in particular climate change (*Hoegh-Guldberg, 1999*; *Hughes et al., 2003*; *Hughes et al., 2017*; *IPCC, 2014*). Therefore, a better understanding of coral holobiont function is critical in order to mitigate strategies to conserve coral reef ecosystems. To this end, the sea anemone Aiptasia (*sensu Exaiptasia pallida*) is becoming a popular model system to investigate the coral-dinoflagellate symbiosis (*Weis et al., 2008*; *Voolstra, 2013*; *Baumgarten et al., 2015*). While recent studies started to look into the association of Aiptasia with *Symbiodinium* (*Thornhill et al., 2013*; *Xiang et al., 2013*; *Hambleton, Guse & Pringle, 2014*; *Wolfowicz et al., 2016*) and bacteria (*Röthig et al., 2016*; *Herrera et al., 2017*), the viral community composition, to our knowledge, has not yet been investigated.

To provide a first insight into Aiptasia viral community composition, we followed a strategy, previously employed by *Brüwer et al. (2017)* to assess *Symbiodinium*-associated viruses, to re-analyze an existing RNA-Seq dataset (*Baumgarten et al., 2015*). The transcriptomic dataset comprised aposymbiotic Aiptasia of the strain CC7 as well as CC7 Aiptasia partially populated and fully symbiotic with endosymbiotic algae of the strain SSB01 (Clade B1, *Symbiodinium minutum*). Our analysis entailed the removal of anemone host and algal endosymbiont sequences and subsequent taxonomic annotation of remaining sequences to assess viral community composition and to determine whether the symbiotic state potentially affects viral association.

## MATERIAL & METHODS

We used a previously published RNA-Seq dataset (NCBI accessions: SRX757525—adult, aposymbiotic Aiptasia CC7, four replicates; SRX757526—adult Aiptasia CC7 partially populated with *Symbiodinium minutum*, four replicates; SRX757528—adult Aiptasia CC7 fully symbiotic with *Symbiodinium minutum*, four replicates) of Aiptasia strain CC7 (*sensu Exaiptasia pallida*) generated for the purpose of assembling a reference transcriptome

for the Aiptasia CC7 genome (*Baumgarten et al., 2015*). Animal culturing, experimental treatments, RNA extraction, and sequencing are briefly outlined below and reported in detail in *Baumgarten et al. (2015)*.

## Culturing of Aiptasia anemones and experimental treatments

Anemones of the clonal Aiptasia strain CC7 were kept in a circulating artificial seawater system at the following rearing conditions: $\sim$25 °C with 20–40 $\mu$mol photons m$^{-2}$ s$^{-1}$ photosynthetically active radiation on a 12 h:12 h light:dark cycle. Anemones were fed freshly hatched *Artemia salina* nauplii twice a week. In order to generate aposymbiotic anemones (i.e., without dinoflagellate endosymbionts), anemones were repeatedly treated with a cold-shock via transfer for 4 h to 4 °C water and subsequent exposure to the photosynthesis inhibitor diuron (Sigma-Aldrich #D2425) at 50 $\mu$M. Anemones were maintained for $\geq$1 month in the above-detailed rearing conditions to ensure absence of repopulation by any residual dinoflagellates. Anemones were inspected individually via fluorescence stereomicroscopy to confirm absence of *Symbiodinium*. To generate partially populated and fully symbiotic anemones, aposymbiotic animals were kept in autoclaved and sterile-filtered artificial seawater (AFSW; other conditions as described above) and infected with strain SSB01 (clade B1, *Symbiodinium minutum*) according to the following treatment: day 1, algae were added at $\sim$10$^5$ cells/ml; day 2, brine shrimp were added without a water change or addition of algae; day 3, AFSW was changed and algae were added at $\sim$10$^5$ cells/ml; day 11, the AFSW was changed. Samples were taken at the mid-point of the 12-h light period on day 0 (aposymbiotic), day 12 (partially populated), and day 30 (fully symbiotic).

## RNA extraction and sequencing

Total RNA was extracted from aposymbiotic, partially populated, and fully symbiotic anemones (see above) using TRIzol (Life Technologies #15596-026; Life Technologies, Carlsbad, CA, USA) following the manufacturer's instructions. The mRNA was extracted from total RNA using Dynabeads oligo(dT)$_{25}$ (Ambion #61002; Ambion, Foster City, CA, USA). Quantity and quality of mRNA were assessed and monitored using a Bioanalyzer 2100 (RNA Nano/Pico Chip; Agilent Technologies, Santa Clara, CA, USA). Subsequent library preparations were conducted using the NEBNext Ultra Directional RNA Library Prep Kit (NEB #E7420; New England Biolabs, Ipswich, MA, USA) with a 180-bp insert size. Libraries were sequenced together on one lane of an Illumina HiSeq2000 sequencer with read lengths of 2 $\times$ 101 bp.

## Sequence data filtering

The software trimmomatic (*Bolger, Lohse & Usadel, 2014*) was used for quality control and read trimming (settings: LEADING:30 TRAILING:30 SLIDINGWINDOW:4:30 MINLEN:35 HEADCROP:6 -phred33). Single reads of paired-end read pairs resulting from quality control were discarded and not considered for downstream analyses. Sequencing adapters were removed with fastq-mcf (*Aronesty, 2011*) (settings: -l 35 –qual-mean 25). The BBsplit script from BBmap v35 (*Bushnell, 2016*) was utilized to remove spiked-in PhiX174 Illumina control sequences (NCBI accession: NC_001422.1), sequences mapping

to the genomes of Aiptasia CC7 (NCBI accession: GCA_001417965.1) (*Baumgarten et al., 2015*; *Liew, Aranda & Voolstra, 2016*) and *Symbiodinium minutum* (NCBI accession: GCA_000507305.1) (*Shoguchi et al., 2013*), as well as any sequences matching 28S rRNAs of sea anemones from the NCBI 'nr' database (version from 16.03.2017; search term: "(((28S AND "cnidarians"[porgn:__txid6073]) AND "anthozoans" [porgn:__txid6101]) AND "sea anemones" [porgn:__txid6103])") (BBsplit settings: minid = 0.7 local = t qin = 33). The reason for the 28S rRNA removal lies in their apparent similarity to two *Baculoviridae*, namely *Choristoneura occidentalis granulovirus* (CLARK taxonomic id: 364745) and *Chrysodeixis chalcites nucleopolyhedrovirus* (CLARK taxonomic id: 320432). Retained sequence reads were used for all subsequent analyses. An overview of filters applied and commands used are available as (Fig. S1, Data S1).

## Viral assemblage analysis

Of the retained sequence reads (see above), only paired reads were considered and annotated to the highest possible phylogenetic level using the classify_metagenome.sh script of CLARK (*Ounit et al., 2015*) (settings: -m 0; remaining settings: default) using NCBI's RefSeq database for bacteria, archaea, and viruses (Data S1). Of note, retained sequence reads were not assembled prior to classification. The database was downloaded using the implemented set_target.sh script (version 1.2.3; default settings; RefSeq release 81). Prior to normalization, viruses that were only annotated with one sequence in one sample (i.e., singletons) as well as read pairs annotating to *Choristoneura occidentalis granulovirus* (NCBI id: NC_008168.1; CLARK taxonomic id: 364745) were removed (see above). In order to correct for differences in sequencing depths across different samples, retrieved sequence counts were normalized using the cumulative-sum scaling (CSS) method implemented in the R Bioconductor package metagenomeSeq (v 1.17.0) (*Gentleman et al., 2004*; *Paulson et al., 2013*; *Paulson, 2014*; *R Core Team, 2016*), and we subsequently only considered sequences that were classified as of viral origin. Information on diverse groups of viruses (i.e., single strand positive sense RNA ssRNA(+), single strand negative sense RNA ssRNA(−), double strand DNA dsDNA, double strand RNA dsRNA, reverse transcribing RNA ssRNA(rt)) as well as known virus hosts (bacteria, fungi, invertebrates, vertebrates, plants, protozoans) were retrieved from either the ICTV website at http://talk.ictvonline.org (*Davison, 2017*) or from ViralZone at http://viralzone.expasy.org (*Hulo et al., 2011*) (version from 11.11.2017) based on viral family or the available next higher phylogenetic assignment. Viral species richness, evenness, and Shannon–Wiener Index (alpha diversity) were estimated using the R package vegan (v. 2.4–2) (*Oksanen et al., 2017*). The R package ggplot2 was used for visualizing the relative abundance of viral taxa and viral families (*Wickham, 2016*).

In order to test for statistical differences in the composition of the viral assemblage of aposymbiotic, partially populated, and fully symbiotic Aiptasia, we conducted analysis of variance (ANOVA) on Pielou's evenness and Shannon–Wiener diversity. Further, we tested for significant differences in relative abundance of viral taxa across conditions. To do this, we tested viral taxa ($n = 116$) with an ANOVA and a posthoc Tukey test (*R Core Team, 2016*) using $p < 0.05$ as a cutoff.
To determine viromes of aposymbiotic, partially populated, and fully symbiotic Aiptasia, we determined all viral taxa that were present in all four replicates (100%) of the respective condition. Those viral taxa that were present in 100% of all aposymbiotic, partially populated, and fully symbiotic Aiptasia samples were considered to be core virome members. The different viromes, including the core virome, were visualized using BioVenn (*Hulsen, De Vlieg & Alkema, 2008*).

# RESULTS

## Viral sequence annotation

A total of 297,207,704 sequence reads (i.e., 148,603,852 paired-end read pairs) distributed over 12 samples, i.e., four replicates of adult Aiptasia anemones across each of three symbiotic stages (aposymbiotic, partially populated, and fully symbiotic) were available for viral sequence annotation (Table 1, Fig. S1). Of those, 262,252,332 (88.24%) sequence reads were retained after quality control, read trimming, and adapter removal. After removal of anemone host, algal endosymbionts, and miscellaneous other sequences (see 'Material & Methods'), 8,597,604 (2.89%) sequence reads were available and used for bacterial, archaeal, and viral annotation using the CLARK classification tool (*Ounit et al., 2015*). A total of 38,090 CLARK-classified sequences were retrieved, of which 90.97% (34,649 sequences) were of bacterial, a smaller fraction of 0.39% (148 sequences) of archaeal, and 8.65% (3,293 sequences) of viral origin. The virus-classified sequences comprised 116 distinct taxa covering 40 viral families (Table S1).

## Aiptasia viral community composition

Aiptasia was associated with a diverse viral assemblage featuring an average species richness of 36.72 (SD $\pm$ 2.98) following *Hurlbert (1971)*. The viral assemblage was evenly distributed as highlighted by an average Pielou's evenness of 0.90 (SD $\pm$ 0.02) and Shannon–Wiener diversity was 3.75 (SD $\pm$ 0.17) across samples (Table 2). Measures of community composition were stable across aposymbiotic, partially populated, and fully symbiotic anemones, as neither Pielou's evenness (ANOVA, $p > 0.88$) nor Shannon–Wiener diversity (ANOVA, $p > 0.50$) were significantly different between different symbiotic states. Almost half of the viral assemblage was encompassed by ssRNA(+) viruses, about a third were annotated as dsDNA viruses, and less than a fifth of the assemblage was comprised by dsRNA viruses. Conversely, ssRNA(−) and ssRNA(rt) were detected at very low frequencies. The ten most abundant viral families accounted for about two-thirds of the viral assemblage (Fig. 1). The most abundant viral families included the *Herpesviridae* (12.00% $\pm$ 0.49%), *Partitiviridae* (9.93% $\pm$ 0.30%), and *Picornaviridae* (9.87% $\pm$ 0.45%). Generally, the assemblage comprised few abundant and many rare viral species across treatments (Fig. 1). The most abundant viral taxon, *Dulcamara mottle virus* (7.16% $\pm$ 0.41%), is a plant-infecting virus of the *Tymoviridae* family and belonged to the fourth most abundant viral family. The next most abundant viral taxa were *Caviid betaherpesvirus* 2 (6.48% $\pm$ 0.29%), *Murid betaherpesvirus* 8 (4.34% $\pm$ 0.28%), *Jingmen tick virus* (4.31% $\pm$ 0.22%), and *Bidens mottle virus* (4.15% $\pm$ 0.23%).

**Table 1 Sequence data overview and read-based annotation.** Numbers of raw and retained (i.e., after quality filtering, trimming, and removal of host anemones, symbiont algae, PhiX, 28S rRNA) sequence reads, as well as number of annotated read pairs are provided. Retained sequence reads were used for taxonomic analysis.

| Condition | Sample | Raw reads | Retained reads | Classified read pairs (total) | Classified read pairs (virus) | Classified read pairs (bacteria) | Classified read pairs (archaea) |
|---|---|---|---|---|---|---|---|
| Apo | R1 | 23,314,626 | 633,310 | 2,220 | 82 | 2,136 | 2 |
| | R2 | 21,623,164 | 640,332 | 2,176 | 203 | 1,965 | 8 |
| | R3 | 23,905,820 | 702,856 | 3,413 | 199 | 3,206 | 8 |
| | R4 | 23,200,990 | 803,114 | 8,407 | 552 | 7,840 | 15 |
| Partial | R1 | 21,485,094 | 798,846 | 8,752 | 733 | 7,980 | 39 |
| | R2 | 23,355,938 | 657,924 | 2,215 | 232 | 1,973 | 10 |
| | R3 | 26,458,678 | 665,100 | 2,318 | 207 | 2,099 | 12 |
| | R4 | 33,532,640 | 818,942 | 2,743 | 277 | 2,452 | 14 |
| Symbiotic | R1 | 23,292,594 | 653,802 | 1,172 | 171 | 996 | 5 |
| | R2 | 25,013,812 | 684,102 | 1,516 | 220 | 1,286 | 10 |
| | R3 | 24,218,018 | 704,760 | 1,284 | 165 | 1,112 | 7 |
| | R4 | 27,806,330 | 834,516 | 1,874 | 252 | 1,604 | 18 |
| Total | | 297,207,704 | 8,597,604 | 38,090 | 3,293 | 34,649 | 148 |
| Percentage | | | | | 8.65% | 90.97% | 0.39% |

Notes.
Apo, aposymbiotic; Partial, partially populated (after 12 days of infection); Symbiotic, fully symbiotic (fully infected, after 30 days of infection); R1–R4, replicated anemones.

**Table 2 Overview of Aiptasia viral community richness, percentage of most abundant viral taxon, evenness, and diversity.** Species richness was estimated following *Hurlbert (1971)* after rarefying to the lowest number of viral-annotated sequences ($n = 82$).

| Condition | Replicate | Species richness (Hurlbert) | Most abundant viral taxon | Evenness (Pielou) | Shannon-Wiener diversity index |
|---|---|---|---|---|---|
| Apo | R1 | 29.264 | 15.85% | 0.911 | 3.312 |
| | R2 | 36.623 | 8.37% | 0.908 | 3.749 |
| | R3 | 37.751 | 8.04% | 0.914 | 3.803 |
| | R4 | 36.821 | 7.07% | 0.863 | 3.792 |
| Partial | R1 | 35.387 | 9.41% | 0.853 | 3.727 |
| | R2 | 35.039 | 10.78% | 0.890 | 3.660 |
| | R3 | 39.679 | 7.73% | 0.921 | 3.900 |
| | R4 | 39.386 | 7.22% | 0.901 | 3.902 |
| Symbiotic | R1 | 34.496 | 10.53% | 0.877 | 3.606 |
| | R2 | 37.559 | 10.00% | 0.903 | 3.784 |
| | R3 | 38.547 | 10.91% | 0.905 | 3.820 |
| | R4 | 40.108 | 8.73% | 0.904 | 3.915 |

Notes.
Apo, aposymbiotic; Partial, partially populated (after 12 days of infection); Symbiotic, fully symbiotic (fully infected, after 30 days of infection); R1–R4, replicated anemones.

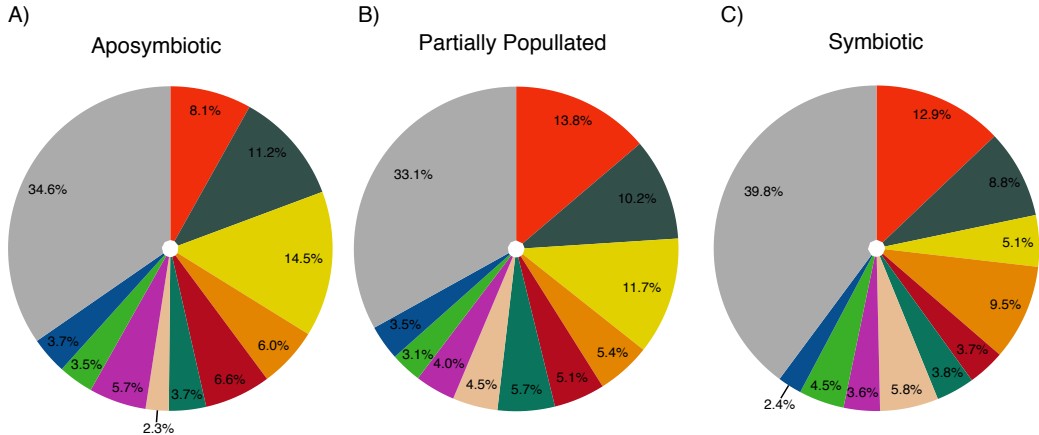

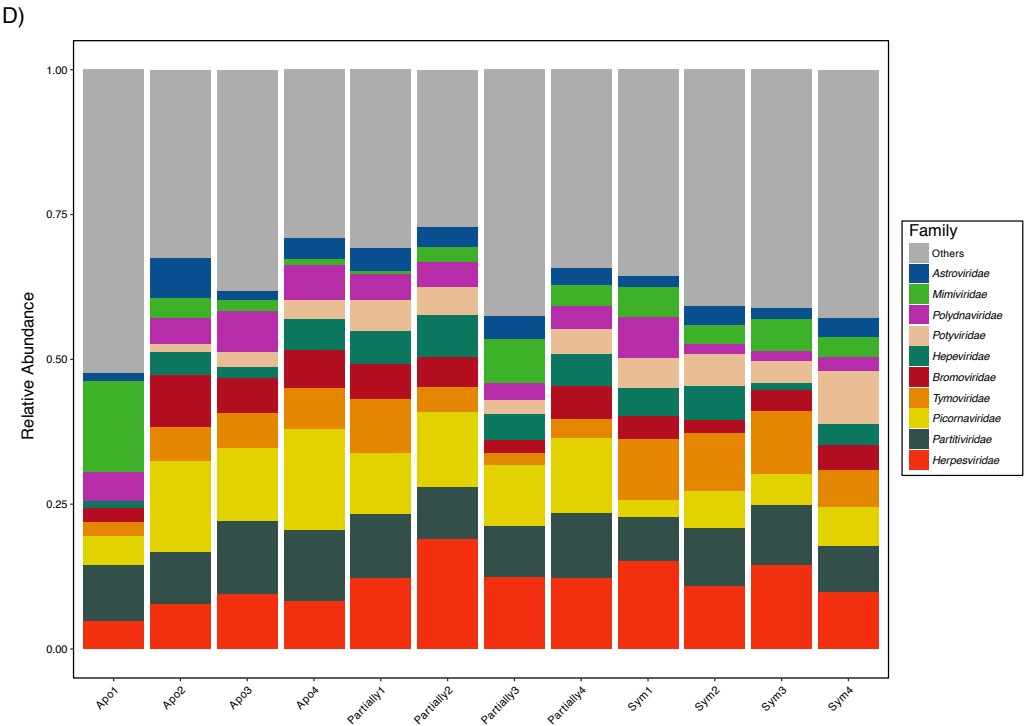

**Figure 1 Aiptasia viral community composition.** Shown are the 10 most abundant viral families associated with adult Aiptasia anemones across three symbiotic stages: (A) aposymbiotic, (B) partially populated, and (C) fully symbiotic; remaining viruses are associated under 'Others'. (D) Shown are the 10 most abundant viral families associated with adult Aiptasia across replicated anemones (1–4). Apo, aposymbiotic; Partially, partially populated (after 12 days of infection); Sym, fully symbiotic (fully infected, after 30 days of infection).

### Viral assemblages of fully symbiotic Aiptasia are different from viral assemblages of aposymbiotic and partially populated sea anemones

Despite the overall similarities in viral assemblage composition, we were interested to assess whether some viral taxa were differentially abundant between symbiotic states/conditions. To do this, we sorted viral taxa according to the condition they were found most abundant in and tested for statistical significance (see 'Material & Methods').

The "aposymbiotic" group included 39 viruses that were found most abundant in aposymbiotic Aiptasia (Fig. 2A), but none were significantly differentially abundant in comparison to partially populated or symbiotic Aiptasia. This group was dominated by *Partitiviridae* and *Picornaviridae* and contained 13 viral taxa with a potential plant host, as well as 11 viral taxa previously described to infect vertebrate hosts, amongst others (Table S2). Interestingly, this group contained five viral taxa that were not identified in the symbiotic condition (indicated by the blue stars, Fig. 2A), as well as two taxa that were not detected in the partially populated Aiptasia (indicated by the yellow stars, Fig. 2A). Of note, the *Anomala cuprea entomopoxvirus* was neither detected in the partially populated, nor the symbiotic condition.

The "partially populated" group consisted of 39 viral taxa that were found most abundant in partially populated Aiptasia (Fig. 2B). Of these, four were significantly differentially abundant in comparison to symbiotic and/or aposymbiotic Aiptasia (indicated by asterisks, Fig. 2B). This group comprised, amongst others, 12 putative plant-infecting viruses and 20 viral taxa previously described to infect vertebrates. A total of eight viral taxa were not detected in both remaining conditions (indicated by the blue and red star, Fig. 2B) and four viral taxa were not detected in at least one of them.

The "symbiotic" group comprised 38 viral taxa that were found most abundant in fully symbiotic Aiptasia. Of these, 11 were significantly differentially abundant in comparison to the aposymbiotic and/or partially populated Aiptasia (Fig. 2C). Further, 11 viral taxa were not detected in either the partially populated or aposymbiotic Aiptasia and six viral taxa were not present in both of the remaining conditions. This group comprised 10 potential plant-infecting viruses, as well as 17 viral taxa previously described to prey on vertebrates, amongst other viruses.

Taken together, 15 of the 116 viral species changed significantly (ANOVA, $p < 0.05$) in relative abundance across conditions (Fig. 2, Table S2). In addition, 23 viral species were not detected in the aposymbiotic, 10 not in the partially populated, and 13 not in the symbiotic anemones. Thus, although the overall viral assemblage was largely consistent with regard to composition (Fig. 1) and abundance (Fig. 2), some viral taxa displayed condition-specific abundance patterns or were specifically present or absent in some conditions, but not in others.

### The Aiptasia core virome

Despite the overall similarities in viral assemblages (Fig. 1) and abundance (Fig. 2) across symbiotic states, we were interested to assess the viromes associated with aposymbiotic, partially populated, and fully symbiotic Aiptasia. To do this, we determined all viral taxa that were 100% present in all four replicates of each respective condition. Partially

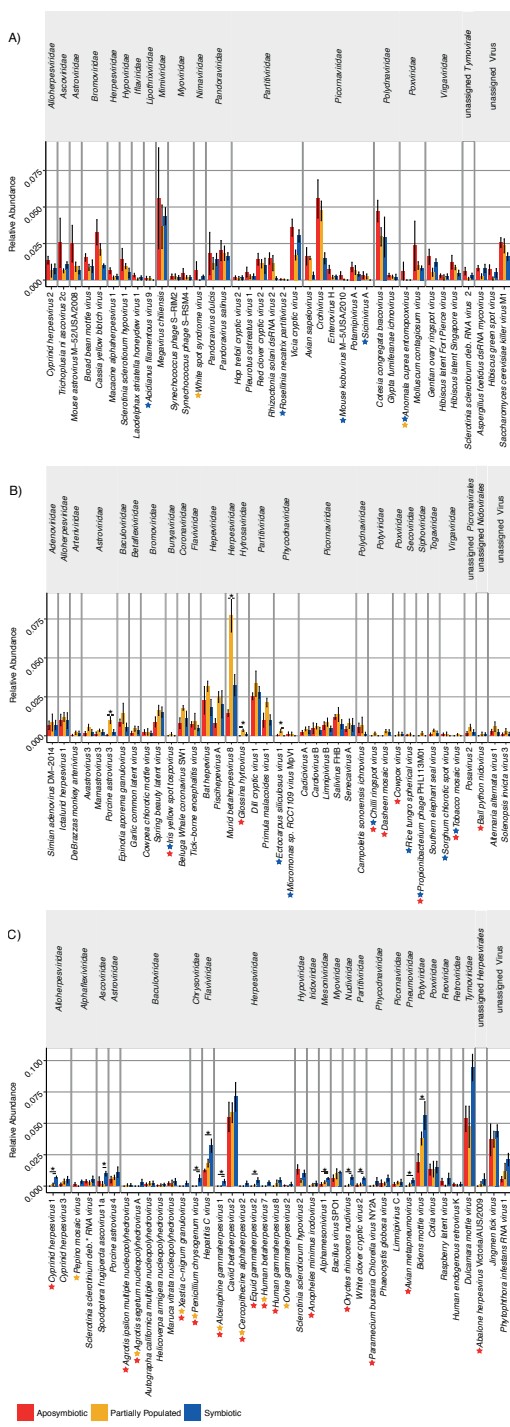

**Figure 2  Relative abundance changes of viruses associated with Aiptasia in relation to aposymbiotic, partially populated, and fully symbiotic anemones.** Viral taxa were separated into three groups according to their abundance across symbiotic states: (A) Viral taxa most abundant in aposymbiotic Aiptasia, (B) viral taxa most abundant in partially populated Aiptasia, and (continued on next page...)

**Figure 2 (…continued)**
(C) viral taxa most abundant in fully symbiotic Aiptasia. Taxa within each group were clustered according to their respective family association (denoted above the graph). Relative highest abundance of each viral taxa across conditions is denoted as follows: red = aposymbiotic condition, yellow = partially populated condition, blue = fully symbiotic condition. Colored stars next to viral taxa names indicate respective absence of viral taxa across different conditions (red = absent in aposymbiotic condition, yellow = absent in partially populated condition, blue = absent in fully symbiotic condition). Asterisks above bars indicate significant differential abundances (ANOVA, $p < 0.05$) and small horizontal lines below indicate which comparisons were significantly different. Abundance estimates are available in Table S1, statistical test details are available in Table S2. deb. = debilitation-associated.

populated Aiptasia anemones harbored the most diverse virome consisting of 41 viral species, followed by the fully symbiotic (32 viral species), and aposymbiotic virome (27 viral species) (Table S3). Thus, consistent with a significant increase in relative abundance for some viral taxa in fully symbiotic anemones, we also found an overall increase in viral diversity. Only few viral taxa were exclusively present in one of the symbiotic states and the majority of viral taxa were present in more than one symbiotic state (Fig. 3). Further, a total of 15 viral taxa across 11 families comprised the Aiptasia core virome (i.e., viral taxa present in 100% of all samples) (Fig. 3, Table S3). The Aiptasia core virome included the four most abundant viral taxa and families, including viruses from the *Herpesviridae*, *Partitiviridae*, *Picornaviridae*, and *Tymoviridae* families.

## DISCUSSION

Despite the importance of microorganisms to their multicellular hosts (*McFall-Ngai et al., 2013*), basic knowledge about the viral community of many organisms, including the model metaorganism Aiptasia, is still lacking. The vastness of next-generation sequencing datasets provides an opportunity to begin to investigate the viral diversity (*Li et al., 2015*), using approaches that filter the host organism and classify remaining sequence reads (*Brüwer et al., 2017*). In this study, we used a previously generated Aiptasia RNA-Seq dataset to gain a first insight into the viral community associated with Aiptasia across three different symbiotic states (aposymbiotic, partially populated, fully symbiotic) with *Symbiodinium*.

It is important to note that the analyzed RNA-Seq libraries were sequenced with the primary scope to produce a reference transcriptome and to complement gene-calling efforts of the Aiptasia genome (*Baumgarten et al., 2015*), and not for a virome analysis. As such, methods for virus enrichment, such as size-based filtrations, the use of cesium chloride (CsCl) gradients, or iron coagulation (*Zhu, Clifford & Chellam, 2005*; *Weynberg et al., 2014*) that would result in an increased yield of viral sequences, and thus, in a potentially more complete representation of the viral community present were omitted. On the other hand, the absence of any size-based filtration allowed for the identification of Nucleocytoplasmic Large DNA viruses (NCLDV), that contain the giant viruses, as recently shown for RNA-Seq based *Symbiodinium* virus expression analysis (*Levin et al., 2017*). NCLDVs are frequently found associated with anthozoans (*Vega Thurber et al., 2017*). Similar to bacteriophages, they are dsDNA viruses (*Hulo et al., 2011*; *Koonin, Dolja & Krupovic, 2015*). As such, they would need to be actively expressed to be captured by RNA-Seq. As a consequence, NCLDVs and bacteriophages are expected to be underrepresented in RNA-Seq libraries.

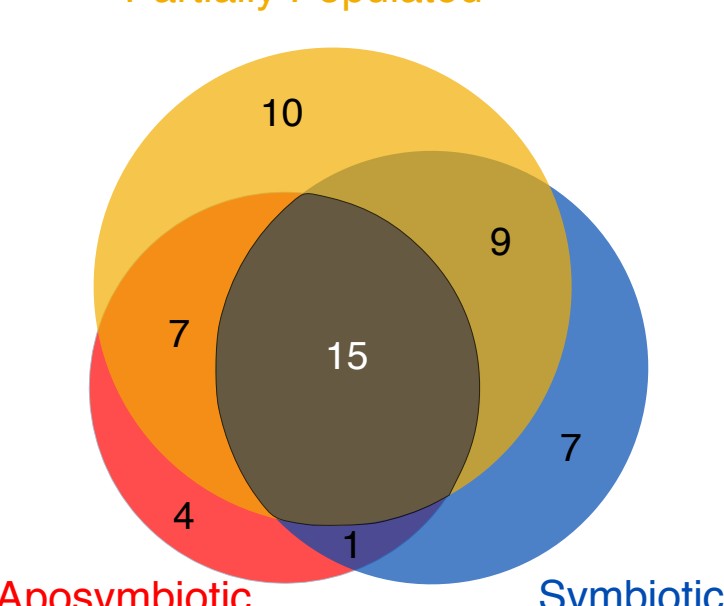

**Figure 3  Viromes associated with aposymbiotic, partially populated, and fully symbiotic Aiptasia.** All viral taxa present in 100% across all four replicates of the respective state (i.e., aposymbiotic (red area), partially populated (yellow area), and fully symbiotic (blue area)) were considered virome members. The core virome (dark gray area) denotes the intersection of viromes from aposymbiotic, partially populated, and fully symbiotic anemones: 15 viral taxa were present in 100% of all samples and are proposed members of the Aiptasia core virome. The areas correspond proportionally to the number of viral taxa they encompass.

Furthermore, the here-analyzed dataset was oligo(dT)-selected prior to sequencing library generation. This resulted in an increase of polyadenylated sequences, which in turn putatively increased our ability to detect ssRNA(+) viruses that contain polyadenylated viral genomes (*Adams, Antoniw & Beaudoin, 2005*; *LeGall et al., 2008*) as well as ssRNA(+) and some dsDNA viruses that polyadenylate their mRNAs (*Majerciak et al., 2013*; *Te Velthuis & Fodor, 2016*). Given that we removed all sequences with similarity to Aiptasia and *Symbiodinium*, we could also not analyze retroviruses that incorporate their DNA into the respective host genomes.

In addition, it should be noted that estimates of relative abundances are based on normalization to sequencing depth, but weren't corrected for genome size differences of the identified viral taxa. Hence, careful consideration has to be applied when assessing the differences in relative abundance across viral taxa. The yield of retained reads was ~5-fold less compared to *Brüwer et al. (2017)*, which may be due to the mapping to two reference genomes, as well as usage of different RNA extraction kits. The TRIzol RNA extraction kit (used in this study) was reported to have a lower efficiency on viral extractions compared to similar RNA extraction kits (*Li et al., 2015*). Similarly, the applied chloroform addition has varying effects on viruses (especially bacteriophages), mainly depending on their lipid

content in the capsid (*Calendar, 2006*), and has previously been shown to cause a decrease in the amount of detectable viral diversity in corals (*Weynberg et al., 2014*). Our analysis therefore provides a first insight into the viral community associated with Aiptasia, rather than a complete characterization, and our efforts should be verified and complemented by viral-targeted metagenomic approaches.

Despite these limitations, based on our analysis, Aiptasia CC7 anemones harbor a diverse viral community that appears to be similar in taxon richness compared to other cnidarians, e.g., *Hydra* (*Grasis et al., 2014*). The here-assessed Aiptasia virome consists of 116 viral taxa from 40 viral families. Interestingly, almost all of the detected viral families have been described in corals (*Wood-Charlson et al., 2015*) or *Symbiodinium* (*Brüwer et al., 2017*). More specifically, 27 (in the case of corals) and 32 (in the case of *Symbiodinium*) out of 40 detected viral families in Aiptasia in this study were previously described. Firstly, this lends further support that RNA-Seq data can be queried to gain a first insight into viral diversity. Secondly, it supports the notion that Aiptasia is a suitable model for the study of cnidarian-dinoflagellate symbiosis, not only at the level of host and algal symbiont biology (*Baumgarten et al., 2015*), but also at the level of bacteria (*Röthig et al., 2016*; *Herrera et al., 2017*) and viruses (this study). It should be noted, however, that *Phycodnaviridae* and bacteriophages of the *Caudovirales* order, which are frequently found associated with corals and *Symbiodinium*, occurred only at very low abundances in the here-analyzed data (*Wood-Charlson et al., 2015*; *Correa et al., 2016*; *Vega Thurber et al., 2017*, but see also *Brüwer et al., 2017*). Importantly, *Phycodnaviridae* and *Caudovirales* are dsDNA viruses. Thus, their paucity in our dataset might stem from biases in the underlying methodology (as discussed above).

The viral assemblages associated with Aiptasia were dominated by *Herpesviridae* (vertebrate-infecting), *Partitiviridae* (plant-, fungi-, and protist-infecting), and *Picornaviridae* (vertebrate-infecting) (*Hulo et al., 2011*) (Fig. 1), which is of particular notice, given that Aiptasia is an invertebrate. However, vertebrate viruses have been frequently found in cnidarian viromes (*Grasis et al., 2014*; *Wood-Charlson et al., 2015*; *Vega Thurber et al., 2017*) and have been described in *Symbiodinium* (*Brüwer et al., 2017*; *Weynberg et al., 2017b*). In a case study on the freshwater polyp *Hydra*, *Grasis et al. (2014)* suggested that the increased vertebrate-virus abundance might be due to a variety of ancestral genes that have been lost in other invertebrates, such as *Drosophila melanogaster* and *Caenorhabditis elegans*, as well as a great similarity of the genome organization. Despite these evolutionary considerations, caution has to be applied when categorizing viruses as vertebrate-, invertebrate-, or fungi-infecting, etc. as such categorization is based on previous research and the information available in databases, which might be biased towards viruses infecting organisms of high economic value, such as crops, livestock, humans, etc. (*Simmonds et al., 2017*). Thus, uneven presentation of viruses from different host organisms in viral databases might further contribute to uncertainties regarding such categorizations.

Besides these uncertainties, viruses similar to the *Herpesviridae* family have been described in many studies investigating anthozoans (*Grasis et al., 2014*; *Wood-Charlson et al., 2015*; *Vega Thurber et al., 2017*), although a recent study describing the DNA and RNA

viromes of seven coral species in the Great Barrier Reef detected only very few *Herpesviridae* (*Weynberg et al., 2017a*). Besides their association with anthozoans, *Herpesviridae* were found associated with *Symbiodinium* (*Brüwer et al., 2017*; *Weynberg et al., 2017b*), which might contribute to the notion that *Herpesviridae*-like viruses were the most abundant viral family in Aiptasia viromes.

The stable association of *Partitiviridae* is of particular interest since their presence in aposymbiotic Aiptasia excludes *Symbiodinium*, suggesting fungi or protists as a potential host, and thus, associates of the Aiptasia metaorganism. Coral fungi have been described as disease-causing agents or secondary scavengers after initial coral holobiont insult (*Ainsworth, Fordyce & Camp, 2017*). Besides such reported detrimental effects, marine fungi may carry out beneficial functions, such as nitrogen fixation or decreasing the impact of UV radiation (among others) in cnidarian metaorganisms (*Ainsworth, Fordyce & Camp, 2017*), but only few studies of coral-associated fungi are available.

*Picornavirales* (including the family of *Picornaviridae*—the third most abundant Aiptasia-associated viral family) have been described as dominant members of the viral community in tropical coastal waters (*Culley et al., 2014*). Thus, it might be less surprising that closely related viral taxa may be associated with the tropical sea anemone Aiptasia, although it might caution the specificity of this association.

Despite an overall diverse and stable viral assemblage associated with Aiptasia, we were interested to further assess whether viral association is different under different symbiotic states (i.e., aposymbiotic, partially populated, and fully symbiotic). This would further contribute to our understanding of the intricacies of the cnidarian-dinoflagellate symbiosis (*Mies et al., 2017*) and provide putative important detail concerning the role of viruses in this symbiosis. We found that distinct viral taxa are specifically present/absent or change in abundance across symbiotic states. Most noticeably, we found significant abundance increases of eleven viral taxa when the host animal becomes fully infected with *Symbiodinium*. We hypothesized that these taxa would be dominated by plant-infecting viruses, given that *Symbiodinium* may come associated with its own distinct set of viruses (*Lawson et al., 2017*). Indeed, we observed plant-infecting *Tymoviridae* and *Portyviridae*, but also vertebrate-infecting *Herpesviridae* in the symbiotic condition. Unfortunately, to our knowledge there is no study characterizing the virome of *Symbiodinium minutum*, which could support our notion. However, other *Symbiodinium* taxa were found to contain NCLDVs closely related to *Phycodnaviridae* or *Mimiviridae,* as well as *Poty-*, *Picorna-*, and *Herpesviridae* (*Brüwer et al., 2017*; *Weynberg et al., 2017b*; *Levin et al., 2017*). Notably, the latter are amongst the seven most abundant viral families in this study. Interestingly, six of the eleven significantly changing viral taxa were not detected in aposymbiotic Aiptasia (red stars in Fig. 2C), lending further support to our initial hypothesis. Although speculative at this point, we suggest that at least some of these viral members are beneficial for the cnidarian-dinoflagellate symbiosis and, thus, important members of the metaorganism.

To better understand the contribution of the virome to a metaorganism, knowledge about the constantly associated viruses (i.e., viral taxa of the core virome) might provide further clues to their importance and ecological significance. A case study in *Hydra* assessed the viral community composition of four different *Hydra* strains and concluded that the

virome, similar to the microbiome, is species-specific (*Grasis et al., 2014*). The Aiptasia CC7 core virome determined in this study comprised 15 viral species from 11 viral families, which is in line with a recent review by *Vega Thurber et al. (2017)* proposing between nine to 12 viral families as members of a coral core virome. More specifically, viruses of the *Mimiviridae*, *Herpesviridae*, and *Poxviridae* families were suggested to be part of the coral core virome (*Vega Thurber et al., 2017*) and are also present in the Aiptasia core virome. The similarities are noticeable, in particular when considering the limitations of the here-analyzed dataset and the circumstance that the determined core virome is derived from clonal Aiptasia CC7 reared under laboratory conditions, and likely to be different from naturally occurring Aiptasia anemones. Of note, as discussed above, viruses similar to the *Herpesviridae* family have been frequently detected in anthozoans (*Grasis et al., 2014*; *Wood-Charlson et al., 2015*; *Vega Thurber et al., 2017*) including this study, and thus, are most likely important members of the cnidarians metaorganism. Bacteriophages of the order *Caudovirales* (including *Siphoviridae*, *Podoviridae*, and *Myoviridae*) that are most abundant members of the *Hydra* virome (*Grasis et al., 2014*) and frequently present in coral viromes (*Wood-Charlson et al., 2015*; *Vega Thurber et al., 2017*; *Weynberg et al., 2017a*) were, however, absent in the Aiptasia core virome, which may be due to methodological issues discussed above. Taken together, despite differences partly due to biases stemming from our approach to use oligo(dT)-selected RNA-Seq data, the Aiptasia viral assemblage identified here exhibits a comparable complexity and displays similarity in composition compared to other anthozoan core viromes.

## CONCLUSIONS

Although the power and validity of the metaorganism concept receives growing attention, we know little about the viral communities associated with host organisms, including many marine invertebrates. To further complement the resources available for Aiptasia as a coral model organism, we analyzed RNA-Seq data to provide a first insight into the virome associated with aposymbiotic, partially populated, and fully symbiotic Aiptasia of the CC7 strain. We find that Aiptasia is associated with a diverse and stable viral assemblage. Certain viral taxa of this community increase their abundance when aposymbiotic anemones establish a symbiotic relationship with their endosymbiont *Symbiodinium*. Hence, the viral assemblage responds to the symbiosis, suggesting putative functional implications that need to be assessed in future studies. Further, we identified candidate members of the Aiptasia core virome that include viruses from the families *Mimiviridae*, *Herpesviridae*, and *Poxviridae*, resembling the composition of coral core viromes. The Aiptasia model metaorganism may facilitate targeted studies to investigate the ecological importance of viruses within the cnidarian-dinoflagellate endosymbiosis with implications for coral reef health.

### List of abbreviations

| | |
|---|---|
| **AFSW** | sterile-filtered artificial seawater |
| **bp** | base pairs |
| **dsDNA** | double-stranded DNA virus |

| **dsRNA** | double-stranded RNA virus |
|---|---|
| **NCLDV** | Nucleocytoplasmic large DNA virus |
| **RNA-Seq** | RNA-sequencing |
| **rRNA** | ribosomal RNA |
| **ssRNA(+)** | positive-sense single-stranded RNA virus |
| **ssRNA(−)** | negative-sense single-stranded RNA virus |
| **ssRNA(rt)** | reverse-transcribing single-stranded RNA virus |

## ACKNOWLEDGEMENTS

We would like to thank Elisha M. Wood-Charlson and two anonymous reviewers for their comments that helped to greatly improve the manuscript.

### Funding

Jan D. Brüwer was funded by a Visiting Student Research Program (VSRP) fellowship awarded by King Abdullah University of Science and Technology (KAUST). Additional supported was provided by baseline funds from KAUST to Christian R. Voolstra. The funders had no role in study design, data collection and analysis, decision to publish, or preparation of the manuscript.

### Grant Disclosures

The following grant information was disclosed by the authors:
King Abdullah University of Science and Technology (KAUST).

### Competing Interests

The authors declare there are no competing interests.

### Author Contributions

- Jan D. Brüwer conceived and designed the experiments, performed the experiments, analyzed the data, prepared figures and/or tables, authored or reviewed drafts of the paper, approved the final draft.
- Christian R. Voolstra conceived and designed the experiments, analyzed the data, contributed reagents/materials/analysis tools, prepared figures and/or tables, authored or reviewed drafts of the paper, approved the final draft.

### Data Availability

The list of bioinformatics software and commands used are provided in a Supplemental File.

### Supplemental Information

Supplemental information for this article can be found online at http://dx.doi.org/10.7717/peerj.4449#supplemental-information.

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
