# Peer review of "First insight into the viral community of the cnidarian model metaorganism Aiptasia using RNA-Seq data"

_PeerJ, doi:10.7717/peerj.4449_

## Round 0.1 · original submission · Major Revisions

All three reviewers agreed that the findings of this study are important and merit publication, but they also emphasized several methodological concerns relating to biases that may stem from the fact that the original RNA-Seq data were not prepared for viral analysis. Please address these issues in the revision of your manuscript, especially regarding the potential impact of certain steps on members of the virome (e.g., poly-A selection biasing against phage transcripts from the Caudovirales, or the differential effects of chloroform on certain viral groups). Also, since the majority of work in this field has been done on coral viromes, note that the reviewers asked you to add references comparing the results of this study to that prior body of work. Acknowledging the limitations of the study, the light in which the results should be interpreted, and strategies that should be employed in future studies to follow-up this work should make this manuscript suitable for publication in PeerJ.

Reviewer 1 ·

Basic reporting

No comment

Experimental design

No comment

Validity of the findings

This study aims to assess the virome of the model organism Aiptasia using previously published RNA-Seq data using an established pipeline to mine for viral sequences. The authors describe a diverse but stable viral community across different stages of symbiosis.

One important area to address here is that the original RNA-Seq data was not obtained with the goal of targeting the viral fraction of the holobiont. Only a small fraction of the <3% of data, which was not host or endosymbiont, was viral. The inclusion of a discussion of a more targeted approach in the pre-sequencing preparation to capture the entire virome fraction would be appropriate. The absence of bacteriophages and members of the Caudovirales (which should be italicised also) is very surprising and I think this reflects the approach taken to screen for viruses in this dataset. This is really my biggest concern about this study and its findings. Bacteriophages of the Caudovirales, as the authors note here, are commonly the most abundant members of, for example, the coral holobiont, so I think this result (i.e. their complete absence here) is somewhat alarming. If bacteriophages are truly absent from Aiptasia (which I cannot really accept) the final sentence of the discussion is not fully the case either, as phages have been implicated in coral health in terms of controlling pathogenic bacteria. I feel this aspect of this study requires more detailed discussion and further justification of the methodological approach used is needed. Is there any additional data to explain this finding? Are phage signatures present before filtering/binning of eukaryotic sequences?

This approach also potentially misses retrotranscribing viruses that may be integrated into the discarded eukaryotic host genomes. Can the authors include some discussion on this point?

It would be relevant and of interest to include discussion/speculation on who the hosts for Partitiviridae may be within the coral holobiont.

Discussion of the species and clade of Symbiodinium used here in terms of any previous studies of the viruses infecting this alga should be included as this is relevant to the changes observed across stages of symbiosis.

More discussion of the prevalence of giant virus-like sequences detected should also be included based on their detection in previous studies.

Picornaviridae have been found to be abundant and diverse in seawater (see Culley et al 2003) and inclusion of the results of such a study would be relevant here.

Fungi are members of the coral holobiont so in line 297 I don’t believe it is unexpected to observe viruses that target fungal hosts in the data. Additional information on what is known of fungi in the holobiont of Aiptasia or other related invertebrates would help in the discussion.

Comments for the author

Minor comments

Ln 318 – Picornaviridae is misspelt
Ln 353: ‘receives’
Ln 363: Herpesviridae is misspelt

·

Basic reporting

The authors mined previously published RNA-Seq data from Aiptasia with and without symbionts to investigate the viral community associated with this holobiont system, including whether the assemblage changes depending on symbiotic state and/or if the holobiont has a core virome regardless of symbiotic state.

The article is well written, the methods section was easy to follow, and I fully support bioinformatic data mining projects. However, my main concern for this manuscript is that these data sets limit how well the authors can address question they are trying to answer. I highly recommend the authors reconsider the scope of what they are actually able to report, given the limitations of the data sets.

Experimental design

The generation of the RNA-Seq data sets started with a Trizol/chloroform step for total RNA extraction, which was followed by an mRNA selection step to purify it from the total RNA in the samples. Unfortunately, both of these steps interfere with sample processing with respect to viruses. Chloroform does not interact with viruses uniformly, which likely changes how the natural viral assemblage is represented for all downstream sample and data processing. In addition, the top Trizol/chloroform layer is recommended for RNA collection, which greatly reduces the viral community those only active inside a cell. Finally, the mRNA selection step would miss any viral transcripts that do not have a poly-A tail, which include most (if not all?) bacteriophages. The analysis doesn’t return a single phage sequence, even though 90.97% of retrieved sequences from CLARK classification were annotated bacteria.

Except for the poly-A selection, none of this is mentioned in the manuscript...

Given that these data are now limited to select eukaryotic viruses, it would be very interesting to see how the list of viral taxa from this data set compares to a list of euk viruses known to have mRNA-poly-A transcripts.

I am also curious how these data annotate against a CLARK database that includes vertebrate sequences, as a test for possible contamination. Since vertebrate genomes are a bit messy, I realize that this might recruit false positives from the current virus annotations. However, as the authors report several vertebrate viruses, this could be informative and relevant.

Also, given that the relative abundance of viral groups were stable across symbiotic state, I would like to see intra/inter variance values reported. Several of the “within replicate” values in Table 1 were somewhat variable, and it would be nice to know if that was significant or not.

I am not up to speed on RNA-Seq data available for the associated Symbiodinium strain, but the data sets seem incomplete without accounting for the viral assemblage that will be present in those cultures, especially since they were added directly. E.g., there didn’t seem to be any attempt to “wash” the symbionts in virus-free seawater. The freshly hatched nauplii are of less concern, as they are not necessarily long-term member of the holobiont, but they are still a potential inoculation source.

Validity of the findings

Given the caveats in the experimental design, I would recommend revising the results section. Aside from that, I have a several additional comments.
First and foremost, I have reservations about the “stability of the viral community”. This can only be stated for the viral representatives present in these limited data sets. For example… Based on previous Symbiodinium/dinoflagellate virus work, I was very surprised to not see Phycodnaviridae represented. Unfortunately, we cannot know if their absence is a true negative or if those sequences were selected against during sample processing. All of this should be mentioned in the text, including a discussion on phycodnaviruses citing (for example) Correa 2013 and 2016, Weynberg 2017, Wood-Charlson 2015 (most already in ref list)

The authors find several viruses that annotate as Picornavirales, which they list as having vertebrate or plant hosts. I would caution against this. The order Picornavirales is vast and has been difficult to characterize in marine systems. Therefore, assignments to a particular family within the order could be a database issue. See Culley et al 2014 mBIO.

Re: Figure 2. I am not totally sold on this figure. The “increase” category includes representatives that increase between apo/partial but decrease again in the fully symbiotic state, indicating a temporary association. There is not enough evidence to relate this to the establishment of the association, nor can we know if they were only coming from the addition of the symbiont cultures (which are never axenic). Either way, they definitely don’t seem to be present (within the limits of these data) in the established symbiosis, so “increase” is misleading. In addition, B shows “decrease” but none of these are significant. This figure does not show the “other” category, where 2 were reported to have significant trend (Supp Tbl 2 – bold). I would like to see these data visualized – what pattern do they fit?

With respect to reporting, as mentioned above, I recommend the authors reconsider the scope of what they are actually able to report, given the limitations of the data sets. For example, line 273-274 should be restated to reflect that this manuscript “provides a highly selective and incomplete view of the viral community”. I realize that greatly diminishes the potential important of this work, but it is a caveat that should be very transparent.

Comments for the author

Line 223 – should this be Supp Table 2?
Line 254-256 – says “includes the four most” but only 3 are listed

Reviewer 3 ·

Basic reporting

In this article, the authors make an effort to describe for the first time the viruses associated with the emerging sea anemone model, Aiptasia. Although using limited sequences from RNA-Seq data, the viral communities described make a first step into characterizing the virome within the anemone holobiont. The authors found that, although biased based on methodology, the majority of the viruses found were ssRNA(-) matching to many vertebrate and plant viruses. Additionally, most of the described viruses were stable throughout different symbiotic states.

Sufficient background and references were included, and figures and tables contained all pertinent information and were clear.

Although the study contained many drawbacks with the sequencing method, the authors do not adequately address them. For example, the viral read numbers are quite low (<100 in one case). Although I realize the published data set cannot be supplemented, it is necessary to acknowledge the insufficiencies of the study which are pointed out below.

Experimental design

This research does address a particularly relevant and meaningful question in holobiont research, and adequately describes the knowledge gap in the Aiptasia model. However, I have some reservations in methodologies that I'll elaborate on below.

It is unclear whether or not these sequences were ever assembled or if all analyses were done on the 180bp reads. Did all reads map to viruses present in the database? Or were only the reads that did analyzed? If they are just reads and not contigs, are they normalized in any way to the reference genome size when determining abundance?

The virus read pairs are quite low in abundance and may preclude you from being able to do adequate analyses on the viral community. The study cited in the article (Brüwer et al. (2017)) that utilizes this same method, assembled contigs (which was not done here) in addition to mapping reads to known viruses. The low number of reads may indicate a lack of coverage and therefore an inability to assemble contigs, however, this drawback needs to be discussed in the manuscript. This low coverage may also be a reason no phages were seen in this sample set, especially because of how much larger the genomes are for Herpes viruses and the other eukaryotic viruses are compared to phages.

Validity of the findings

Many of the findings are nicely described. The concerns on how the abundances are calculated and the lack of assemblies to infer diversity of the virome are described in the above experimental design section.

However, the identification of a "core" virome in Aiptasia is quite misleading because these animals are all the same species and are all reared together in ASW in the lab. I would describe this as more of a stable virome across symbiotic states.

Comments for the author

The Aiptasia model is a highly tractable and emerging model system in holobiont research, and characterizing the viral component is necessary. This article represents an initial study on the viral community in Aiptasia and the changes and stability in the structure over various symbiotic states. The authors have done as much as possible with the available data, however, it is very important to address the caveats of the study and methodologies which the authors largely failed to do. In particular, low number of reads, the lack of reported assemblies and the absence of bacteriophages within the viral is quite peculiar seeing as there are numerous bacterial reads and that phages are so dominant in Hydra, other corals and marine invertebrates. These may indeed be fault of the RNA-Seq method, but suggesting a future direction and necessity of verifying these findings through metagenomics would be nice to include in the Discussion or Conclusion.

A few suggested basic edits:
In the abstract (lines 28-29), the authors state that only few studies have "characterized viral diversity and the potential underlying functional importance to coral holobionts", and this is just not true. I would suggest rewording to emphasize the limited knowledge of the Aiptasia virome and the use for it as a model for coral research.

Line 61: and immune system? Clarify immune function or immunity
Line 295: Delete extra single parenthesis.
Line 354: "many animals and host" seems redundant
Line 355: coral holobionts are actually one of the most well studied seeing as studying these organisms was how the concept originated. Would be better to focus on anemones and other marine inverts.
Line 365: importance of viruses *within* the ....

Figure 2 and legend extends far beyond the margins.

---

## Round 0.2 · Minor Revisions

Please address the remaining comments raised by Reviewer 2, and then your manuscript should be acceptable for publication.

Reviewer 1 ·

Basic reporting

No comment

Experimental design

The shortcomings of the experimental design have now been well-explained in the revised test and the caveats have been presented in more detail, which was lacking in the original manuscript.

Validity of the findings

The revised manuscript now includes much more detailed explanation on the shortcomings in the original. The discussion now includes the limitations of the approach used and the subsequent data that was analysed.

·

Basic reporting

The revised article reads much smoother and does a much better job representing the data. I only have a few comments.
1. In the abstract, the authors comment on the limitations of the data set by only saying “Despite the limitations of the current approach, our study…” I feel this really leaves the reader hanging. I suggest expanding slightly so that the opening phrase lets readers know where and why the approach is limited. E.g., “Despite the non-random selection of viral genetic material due to the sequencing method used, our study…”
2. I am not sure where viral ecology is in the progression from using “community” to “assemblage” as terms to describe limited snapshots of viral associations with hosts, but I would encourage the authors to consider the terms. I get the sense, only from personal conversations with our field, that things are moving towards “assemblage”.
3. Line 137-139: is there a reference for this statement?
4. Line 238: please be consistent – are they stars (see line 229) or pentagons?
5. Link 289-290: “…NCLDV, also known as giant viruses, …” is an incorrect statement. They contain the giant viruses, but not all families in NCLDV are giant viruses.

Experimental design

no comment

Validity of the findings

I am still not overly convinced by the high abundance of herpes virus sequences, mostly b/c when we did a sequence recruitment to a herpes genome in the databases (SuHV1, NCBI genome NC_006151), a lot of the higher blast score reads were to repeat regions. (Wood-Charlson 2015, Figure 2).

If you explore this, and the reads recruit to annotated regions, then I would be convinced.

Reviewer 3 ·

Basic reporting

The authors use clear and unambiguous language throughout the text and have addressed my concerns. The insufficiencies of the study have been adequately addressed and references updated.

Experimental design

The authors have addressed my concerns on the design of the study and have acknowledged its limitations thoroughly.

Validity of the findings

The authors have addressed my concerns in the description of their findings and it's caveats.

Comments for the author

This paper will no doubt serve as a catalyst for more in depth studies investigating the role of viruses in developing model systems for host-microbe interactions.

---

## Round 0.3 · accepted · Accept

Thanks for taking the time to carefully address the reviewer's comments.